# Oxygen Plasma Technology-Assisted Preparation of Three-Dimensional Reduced Graphene Oxide/Polypyrrole/Strontium Composite Scaffold for Repair of Bone Defects Caused by Osteoporosis

**DOI:** 10.3390/molecules26154451

**Published:** 2021-07-23

**Authors:** Xiaoxue Mai, Zebiao Kang, Na Wang, Xiaoli Qin, Weibo Xie, Fuxiang Song

**Affiliations:** 1School of Stomatology, Lanzhou University, Lanzhou 730000, China; Maixx19@lzu.edu.cn (X.M.); Kangzb19@lzu.edu.cn (Z.K.); wangna20@lzu.edu.cn (N.W.); Qinxl20@lzu.edu.cn (X.Q.); 2Lanzhou University Second Hospital, Lanzhou 730000, China

**Keywords:** strontium (Sr), three-dimensional (3D), scaffold, bone defect, tissue engineering, oxygen plasma technology

## Abstract

Repairs of bone defects caused by osteoporosis have always relied on bone tissue engineering. However, the preparation of composite tissue engineering scaffolds with a three-dimensional (3D) macroporous structure poses huge challenges in achieving osteoconduction and osteoinduction for repairing bone defects caused by osteoporosis. In the current study, a three-dimensional macroporous (150–300 μm) reduced graphene oxide/polypyrrole composite scaffold modified by strontium (Sr) (3D rGO/PPY/Sr) was successfully prepared using the oxygen plasma technology-assisted method, which is simple, safe, and inexpensive. The findings of the MTT assay and AO/EB fluorescence double staining showed that 3D rGO/PPY/Sr has a good biocompatibility and effectively promoted MC3T3-E1 cell proliferation. Furthermore, the ALP assay and alizarin red staining showed that 3D rGO/PPY/Sr increased the expression levels of ALP activity and the formation of calcified nodules. The desirable biocompatibility, osteoconduction, and osteoinduction abilities, assure that the 3D macroporous rGO/PPY/Sr composite scaffold offers promising potential for use in the repair of bone defects caused by osteoporosis in bone tissue engineering.

## 1. Introduction

Previous studies have reported a gradual increase in the incidence of bone defects caused by osteoporosis due to the aging of the population. More than 200 million people are estimated to suffer from osteoporosis worldwide [1]. Furthermore, previous studies have reported that about 50% of people aged 65 suffer from osteoporotic fractures [2]. However, the repair of bone defects caused by osteoporosis has been a challenge for clinical diagnosis and treatment. Previous studies have demonstrated that osteoporosis delays the natural healing process of bone fractures and reduces the formation rate and quality of new bone [3]. Current efforts for the improvement of bone defect repairs for osteoporotic fractures entail the improvement of physical and chemical properties as well as the physiological activity of implant materials to improve osteogenic activity or inhibit osteoclast activity in situ. For instance, Liu et al. [4] injected calcium sulfate into the osteoporotic vertebrae of sheep, which led to a significant improvement in the local bone mineral density and the biomechanical properties of the vertebrae after three months compared with the control group. Moreover, Yang et al. [5] demonstrated improved osteogenesis in osteoporotic rats upon the implantation of strontium-containing calcium sulfate hemihydrate. However, these repair materials lacked the morphology of a natural bone macroporous structure and sufficient biological activity. Scaffolds with three-dimensional (3-D) porous networks provide effective matrix conditions for bone tissue engineering [6]. Previous studies have reported that 3D hierarchical scaffolds with a large pore size of 200–500 um and porous bioactive materials deposited on the backbone of the scaffold could effectively favor tissue growth and capillary formation [7]. Macro-sized pores in scaffolds benefit the transport of nutrients and simultaneously provide a large specific area for tissue growth [8]. A porous nanocomposite coating on the skeleton of the scaffold could markedly enhance the adhesion of osteoblast at early stages of implantation and stimulate the proliferation and differentiation of osteoblast at subsequent stages of implant assimilation [9]. Therefore, there is an urgent need to develop new composite scaffold materials with a macroporous structure and confirm their bioactivity for the repair of bone defects in osteoporotic fractures.

Strontium (Sr), an essential trace element, has been shown to effectively stimulate bone formation and inhibit osteoclast activity [10,11,12]. Previous studies have reported that Sr can regulate the differentiation ability of bone marrow mesenchymal stem cells and enhance their osteogenic differentiation ability, thereby playing an important role in regulating the balance of bone metabolism [13]. In addition, previous studies have reported that Sr can also inhibit the activity of osteoclasts via interference with the mechanism of the closed area of osteoclasts [10]. Consequently, strontium-based anti-osteoporosis drugs such as strontium ranelate have been widely recognized by clinicians and patients [14,15]. However, Sr alone cannot fix and play a role in the repair of bone defects. Several previous studies have reported different composite materials modified by Sr that could be used to repair bone defects caused by osteoporosis. For instance, previous studies have demonstrated the use of strontium-doped hydroxyapatite/silk fibroin (SrHA/SF) biological composite nanospheres as biomaterials for inducing the repair of bone defects [16]. Strontium-modified hydroxyapatite (cSrHA) was also prepared in a previous study, where it was established that Sr has a positive effect on the repair of bone defects caused by osteoporosis [17]. However, these substrate materials lack regular three-dimensional macroporous structures similar to those of natural bone. In a previous study, a three-dimensional macroporous reduced graphene oxide/polypyrrole (3D rGO/PPY) composite scaffold, with a similar structure and morphology to natural bone, was successfully prepared through the layer by layer self-assembly (LBL) method [18]. This scaffold was shown to have good biocompatibility with osteoblasts.

Graphene exhibits several unique properties, including high electron mobility at room temperature, exceptional thermal conductivity, the highest theoretical specific surface area, and superior mechanical properties with a Young’s modulus of 1 TPa. These properties make graphene an ideal material for several applications. In addition, graphene is the material of choice for future applications in the electronics and composites industry [19,20]. Furthermore, graphene has shown good performance in medical applications, such as for cancer treatments [21], disease diagnosis [22], microbiological detection and drug/gene delivery [23], antibacterial materials [24], and biosensors and biological imaging [25]. Many previous studies have shown that graphene and its derivatives can induce the adhesion, proliferation, and differentiation of bone marrow stromal cells (BMSCs) on implants and biological scaffolds [26,27]. Hence, graphene is an excellent potential biocompatible material for medical applications, particularly as a coating for the layers of scaffolds to improve their bioactivity. Polypyrrole (PPy), an electrically conductive polymer, has a good biocompatibility. A previous study reported that extracts of PPy-silicone tubes bridging sciatic nerve gaps in rats showed no evidence of acute or subacute toxicity, pyrogenicity, hemolysis, allergenicity, or mutagenesis, even after 24 weeks of operation in rats [28]. Notably, polypyrrole has also been shown to enhance the adhesion of osteoblasts [29]. Therefore, in addition to being biocompatible, a composite scaffold comprising GO and PPY scaffolds also promotes the proliferation and adhesion of osteoblasts, which has been confirmed in previous studies [18].

Plasma etching is the most common approach used in semiconductor processing for the fabrication of electronic devices. Plasma etching with oxygen is a particularly common tool used to modify the surface properties of materials, such as surface energy, morphology, and resistivity [30]. In addition, oxygen plasma treatment with a high chemical reactivity to carbon is an effective strategy for the modification of carbon-based materials to alter their nano-structures and properties [31]. A previous study reported that oxygen plasma etching on multilayer graphene is isotropic (particles attack water from all angles) [32], and oxygen exposure to graphene and graphite proceeds with the formation of epoxide groups at bridge or top sites on the graphene basal plane by oxygen adsorption [33]. Therefore, oxygen plasma etching provides many active sites for the modification of strontium ions to graphene-based scaffolds after oxygen plasma treatment.

In the current study, oxygen plasma technology was used to fabricate Sr-modified 3D rGO/PPY composite scaffolds. In addition, the morphology, structure, and chemical composition of Sr-modified 3D rGO/PPY composite scaffolds (3D rGO/PPY/Sr) were characterized by SEM, EDS, FTIR, and XRD. Biocompatibility was evaluated by MTT and acridine orange/ethidium bromide (AO/EB) double staining assays in vitro. Osteogenic activity was evaluated by alkaline phosphatase (ALP) activity assays and alizarin red staining experiments in vitro, respectively.

## 2. Results and Discussion

### 2.1. Preparation and Characterization 

Scaffold material for tissue engineering with regular, three-dimensional macroporous structure morphology is beneficial for osteogenesis. The SEM results of the 3D rGO/PPY composite scaffold are shown in Figure 1b–d. The scaffold pore size of 150–300 μm is similar to the optimal pore size (200 μm) for osteoblast growth, which could provide mechanical support for cell growth and tissue regeneration [34]. At higher magnification, the scaffold surface presented small ball-like protrusions, which were PPY particles that formed on the surface of the 3D rGO/PPY scaffold by electrochemical deposition (Figure 1c,d). Moreover, particles were “fused” with each other, thereby increasing the mechanical properties of the base scaffold. The particles can be used as a scaffold material that is easy to maneuver. On the other hand, the rough surface formed by PPY particles has a good biocompatibility and promotes the adhesion and proliferation of osteoblasts. The EDX findings showed that, besides the main C, N, and O elements of the 3D rGO/PPY modified by the Sr scaffold itself, the increased Sr element was uniformly deposited on the skeleton surface of the scaffold (Figure 1e–h). This finding indicated that Sr was successfully modified on the skeleton surface of the 3D rGO/PPY scaffold.

The chemical construction of the scaffolds was characterized by XRD and FTIR. As shown in Figure 1i, a wide diffraction peak can be observed at 21.4° in the 3D rGO/PPY spectra, which is attributed to graphene. This is because the GO had a low crystallinity for its irregular arrays of atoms in the three-dimensional structure. After being modified by Sr, several new peaks appeared, and the peaks at 2θ = 11.2°, 19.6°, 26.8°, 28.8° and 44.3°, corresponded to strontium salt [35,36]. The characterization diffraction peaks of GO were not clearly shown in the 3D rGO/PPY/Sr; this is because the amount of GO is relatively low compared to that of strontium salt. In the FTIR spectra (Figure 1j), the peaks at 3438 cm^−1^, 1543 cm^−1^ and 1032 cm^−1^ indicate that -OH, C=C, and C-O bonds. Vibrational stretching at 2815 cm^−1^ indicates the C–H stretching of the aliphatic alkyl group. These are attributed to graphene oxide [37]. The vibrational stretching at 1458 cm^−1^ indicated the presence of the carboxylate group. The peaks at 894cm^−1^, 660 cm^−1^ and 606 cm^−1^ corresponding to the carbonates (CO_3_^−2^) of the reacted SrCO_3_ were observed [36,38]. It was indicated that Sr was successfully modified on the surface of the 3D rGO/PPY scaffold [39].

### 2.2. MTT Assay

The cytotoxicity of the 3D rGO/PPY and 3D rGO/PPY/Sr composite scaffolds were evaluated using MC3T3-E1 cells with MTT assays at 24, 48, and 96 h (Figure 2). MTT assay findings showed that the OD value of the control group and all experimental groups increased gradually with the co-culture time increasing from 24 to 96 h (Figure 2). Notably, the OD values of 3D rGO/PPY and 3D rGO/PPY/Sr scaffold groups were significantly higher compared with those of the control group (*p* < 0.05). Moreover, the OD value of the 3D rGO/PPY/Sr scaffold group was significantly higher compared with that of the 3D rGO/PPY group at the same co-culture time point (*p* < 0.05). The findings of the MTT assay also show that both the 3D rGO/PPY and 3D rGO/PPY/Sr composite scaffolds were not cytotoxic and could promote the proliferation of MC3T3-E1 cells. Notably, the 3D rGO/PPY/Sr composite scaffold had a stronger capacity to promote MC3T3-E1 cell proliferation compared with the 3D rGO/PPY scaffold. This finding might be related to the modification of Sr, which was also reported in previous studies, an indication that Sr promoted MC3T3-E1 cell proliferation [40].

### 2.3. Acridine Orange/Ethidium Bromide (AO/EB) Double Staining Assay

The apoptosis, adhesion, and proliferation of MC3T3-E1 cells on 3D rGO/PPY and 3D rGO/PPY/Sr composite scaffolds were further evaluated using the AO/EB double staining experiment. Representative live/dead fluorescent images showed that almost all MC3T3-E1 cells, both on the backbone of the 3D rGO/PPY and 3D rGO/PPY/Sr composite scaffolds, glowed green throughout the culture cycle and no red-stained cells were detected (Figure 3). This finding may indirectly indicate that both the 3D rGO/PPY and 3D rGO/PPY/Sr composite scaffolds are not cytotoxic. In addition, with the culture time shifting from 24 to 96 h, the number of MC3T3-E1 cells, both on the 3D rGO/PPY and 3D rGO/PPY/Sr composite scaffolds, also increased gradually. Low numbers of cells were observed in 24 h, whereas more cells were observed in 48 h. High numbers of cells covered almost the entire backbone of the scaffold in 96 h. The number of cells on the 3D rGO/PPY/Sr scaffold group was consistently higher compared with that in the 3D rGO/PPY scaffold group at the same point of culture time. The findings of the AO/EB staining further indicated that both the 3D rGO/PPY and 3D rGO/PPY/Sr composite scaffolds promoted the adhesion and proliferation of MC3T3-E1 cells, although the ability of the 3D rGO/PPY/Sr composite scaffold was higher compared with that of the 3D rGO/PPY scaffold. These findings are consistent with those of the MTT assay and may be attributed to Sr incorporation [40].

### 2.4. ALP and Alizarin Red Staining Experimental

Ability to induce the differentiation of MC3T3-E1 preosteoblasts into osteoblasts by 3D rGO/PPY and 3D rGO/PPY/Sr composite scaffolds was also determined in the current study using ALP activity and the alizarin red staining experiment. Previous studies have reported that the activity level of ALP and the number of calcified nodules reflect their osteogenic differentiation ability [41]. Higher activity levels of ALP and a higher number of calcified nodules correlate with a stronger osteogenic differentiation ability. No significant differences were observed in the ALP activity levels between the 3D rGO/PPY and the control groups at both the 7th and 14th days (*p* > 0.05) (Figure 4). However, ALP activity levels in the 3D rGO/PPY/Sr group were 2 and 1.7 times higher compared to that in the 3D rGO/PPY group at the 7th and 14th days (*p* < 0.05). Furthermore, the findings of the alizarin red staining experiment showed that areas of red dye and numbers of calcified nodules gradually increased both in the 3D rGO/PPY and 3D rGO/PPY/Sr group as the co-culture time gradually increased (Figure 5). In contrast, the areas of red dye and numbers of calcified nodules of the 3D rGO/PPY/Sr group were higher compared with those of the 3D rGO/PPY group at the same point of co-culture time.

The findings of the current study established that the 3D rGO/PPY composite scaffold modified by Sr promotes the expression of ALP activity levels and the formation of calcified nodules. In addition, the current study findings showed that 3D rGO/PPY modified by Sr can potentially improve the osteogenic properties of biomaterials, which may be attributed to the possibility that Sr^2+^ can activate calcium sensing receptors (CaR) and increase osteoprotegerin (OPG) production [42,43].

### 2.5. Morphological Observation 

SEM images indicated that MC3T3-E1 had a good morphology and spreading on scaffolds (Figure 6). After 48 (a) and 96 (b) hours of co-culture, the cells on 3D rGO/PPY/PDA/Sr scaffolds were well developed and spread over the surface with prominent lamellipodia extension.

### 2.6. Mechanically Robust Performance

The findings of the current study indicated that the 3D rGO/PPY/Sr scaffold was loaded with a compression strain up to 90% (Figure 7a); the strain vs. stress curve presented a non-linear elastic deforming mechanism during compression. The highly loaded 3D rGO/PPY/Sr scaffold was compressed into a thin ‘pancake’, with a maximum strength of up to 2.61 kPa corresponding to a compressive strain of 90% (Figure 6a). The compacted 3D rGO/PPY/Sr scaffold then reverted to its original porous status, indicating excellent robustness for a structure undergoing a large-scale deformation. Moreover, as promising bioscaffolds for cell attaching and creeping, 3D rGO/PPY/Sr scaffolds are supposed to have good mechanical elasticity and tough capability to maintain structural stability during cultivation-transfer processes. The original dry and stiff 3D rGO/PPY/Sr scaffolds were moistened in culture solvent and then transferred with the structure remaining tough and robust (Figure 7b). Such superior performance of 3D rGO/PPY/Sr scaffolds in resisting large deformation and tension force-induced shrinkage enables them to be used widely as biocompatible scaffolds. Arbitrary configurations such as the O-ring and other shaped rGO/PPY films can also be easily fabricated using the corresponding shapes of the template (Figure 7c), implying that the configuration of 3D rGO/PPY full-carbon foam only depends on the shape of the template. This ensures the precise design of the preferred configuration of rGO/PPY scaffolds based on the requirement of bone defects, without complicated subsequent reprocessing.

### 2.7. Sr Release Determination

The 0.0015g 3D rGO/PPY/Sr scaffolds were immersed in 5 mL of PBS at 37 °C, and PBS was collected after 1, 2, 4, 8, 12, 24, 48, 72, 96, and 120 h, respectively. PBS containing released Sr was analyzed using ICP−OES. As shown in Figure 8, the cumulative release of Sr^2+^ gradually released from the 3D rGO/PPY/Sr scaffold to almost constant and slow release after 24 h. This is preferable for bone regeneration, as the scaffold can maintain the moderate concentration of Sr^2+^ ions for a long time. 

## 3. Materials and Methods

### 3.1. Fabrication and Characterization of the 3D rGO/PPY/Sr

The 3D rGO/PPY scaffold was prepared based on the previously reported method [18]. Three-dimensional rGO/PPY samples measuring 1 × 2 cm were treated with oxygen plasma for 20 min and immersed in 0.2 M strontium chloride solution for 1 h. The treated 3D rGO/PPY samples were then thoroughly rinsed with deionized water to obtain ultimate composite scaffolds (3D rGO/PPY/Sr). The surface morphology and chemical composition of composite scaffolds were characterized through scanning electron microscopes (SEM, JSM-5601LV) equipped with an Energy Dispersive X-ray spectrometer (EDX), an X-ray powder diffractometer (XRD, D/max-2400, Rigaku, Cu Kα), and a Fourier infrared spectrometer (FT-IR, IFS66V/S, Karlsruhe, Switzerland). Mechanical compressibility was measured using an electronic universal testing machine (AGS-x, Shimadzu, Kyoto City, Japan) with a loading rate of 50 mm min^−1^. The release property of Sr ions from the scaffolds was analyzed using inductively coupled plasma optical emission spectrometry (ICP−OES, Varian, San Diego, CA, USA).

### 3.2. MTT Assay

Osteoblast-like MC3T3-E1 cells (clonal mouse osteoblastic cell line, ATCC, Rockville, MD, USA) were cultured in Dulbecco′s Modified Eagle Medium (DMEM, Gibco, Waltham, Massachusetts, USA) containing 10% fetal bovine serum (FBS, Gibco, Waltham, Massachusetts, USA) at 37 °C, with the medium being changed every two days. The cytotoxicity of the 3D rGO/PPY and 3D rGO/PPY/Sr composite scaffolds were examined using a 3-(4,5-dimethyl-2-thiazolyl)-2,5-diphenyl-2-H-tetrazolium bromide (MTT, St. Louis, MO, USA) assay, and the absorbance of the solution in each well was determined at a wavelength of 490 nm using a microplate reader (Bio-Rad iMark, Hercules, CA, USA).

### 3.3. Acridine Orange/Ethidium Bromide (AO/EB) Double Staining

Reagents AO and EB were purchased from Sigma (St. Louis, MO, USA). Living and apoptotic cells were determined morphologically after AO/EB staining, followed by fluorescence microscopy inspection. Briefly, MC3T3-E1 cells were seeded at a density of 10^6^ cell/mL concentration in a 24-well plate incubated with the 3D rGO/PPY and 3D rGO/PPY/Sr composite scaffolds. After 24, 48, and 96 h, all cultured scaffolds were harvested, washed thrice with PBS, and dehydrated with 4% paraformaldehyde. The scaffolds were then stained with AO/EB solution (1:1 mixture of 1 mg/mL ethidium bromide (EB) and 1 mg/mL acridine orange (AO)) in the dark for 3 min. Cellular morphology was evaluated using a fluorescence microscope (Olympus BX53, Tokyo, Japan).

### 3.4. Alkaline Phosphatase (ALP) Activity 

The differentiation of MC3T3-E1 cells was determined by ALP activity, which was assayed using an Alkaline Phosphatase Assay Kit (Nanjing Jiancheng Bioengineering Institute, Jiangsu, China) based on the manufacturer’s protocol. Briefly, MC3T3-E1 cells were seeded onto samples in 6-well plates at a density of 10^4^ cell/mL concentration. After co-culturing for 7 and 14 days, scaffolds were washed thrice with PBS and lysed with 0.01% Triton-X 100 to dissolve cells. The mixture was centrifuged at 2000 rpm for 15 min. The supernatant was transferred to an EP tube and the ALP working solution was added. After incubation at 37 °C for 15 min, 1 M NaOH was added to terminate the reaction. Absorbance at 520 nm was determined using a microplate absorbance reader (Bio-Rad iMark). All experiments were undertaken in triplicate.

### 3.5. Alizarin Red Staining

The ECM mineralization of MC3T3-E1 co-cultured with the 3D rGO/PPY and 3D rGO/PPY/Sr composite scaffolds were tested through alizarin red staining. MC3T3-E1 cells were seeded onto samples in 6-well plates at a density of 10^4^ cell/mL concentration. After 3 days, the cultured solution was replaced with the osteogenic medium (DMEM supplemented with 10% PBS, 10 mM β-glycerophosphate, and 50 μg/mL ascorbic acid). Cells were rinsed twice with PBS after co-culturing for 7, 14, and 21 days and fixed in 4% paraformaldehyde for 10 to 20 min at room temperature. Cells were stained with alizarin red for 30 min and washed thrice with PBS. The formation of calcium nodules was observed using an inverted microscope.

### 3.6. Morphological Observation

MC3T3-E1 were cultured, grown on scaffolds for 24 and 48 h, and fixed (4% paraformaldehyde) for 15 min. Samples were then rinsed with PBS and dehydrated, followed by drying for 1 min and gold sputtering, before observation using a scanning electron microscope (SEM, JSM-5601LV)

### 3.7. Statistical Analysis

Statistical analyses were performed using SPSS version 12.0. One-way analysis of variance (ANOVA) with Bonferroni post hoc tests was undertaken to examine differences among groups. Values of *p* < 0.05 were considered statistically significant.

## 4. Conclusions

The current study developed a simple and effective modification method to fabricate a 3D rGO/PPY/Sr scaffold using oxygen plasma-assisted strategy. The 3D rGO/PPY/Sr scaffold showed an outstanding biocompatibility and a prominent ability to promote the proliferation of MC3T3-E1 cells. By co-culturing with MC3T3-E1 cells, the 3D rGO/PPY/Sr scaffold showed higher levels of ALP activity and an ability to promote the formation of calcified nodules. To the best of our knowledge, 3D rGO/PPY/Sr composite scaffolds are potential materials for the repair of bone defects caused by osteoporosis in the future.

## Figures and Tables

**Figure 1 molecules-26-04451-f001:**
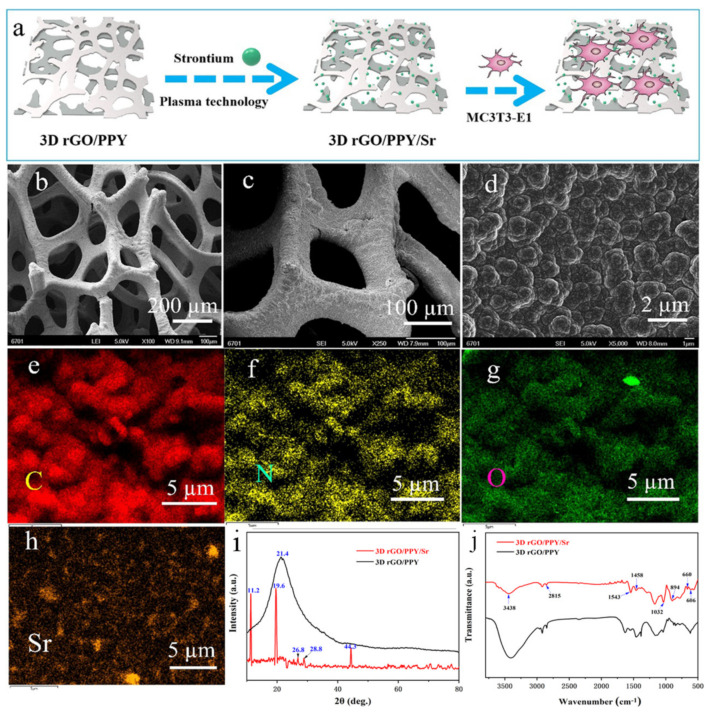
(**a**) Flow diagram of experiment process. (**b**–**d**) SEM images of the 3D rGO/PPY scaffold, (**e**–**h**) the EDX results of the 3D rGO/PPY/Sr scaffold. The XRD (**i**) and FTIR (**j**) results of the 3D rGO/PPY before and after Sr modification, respectively.

**Figure 2 molecules-26-04451-f002:**
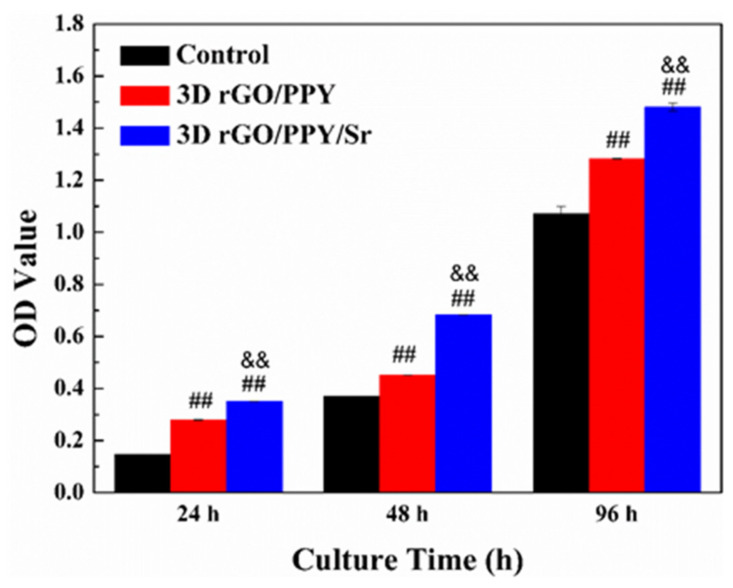
The results of the MTT assay for 3D rGO/PPY and 3D rGO/PPY/Sr scaffolds after co-culture with MC3T3-E1 cells at 24 h, 48 h, and 96 h, respectively. (## *p* < 0.05 3D rGO/PPY vs. control group, && *p* < 0.05 3D rGO/PPY/Sr vs. 3D rGO/PPY group.).

**Figure 3 molecules-26-04451-f003:**
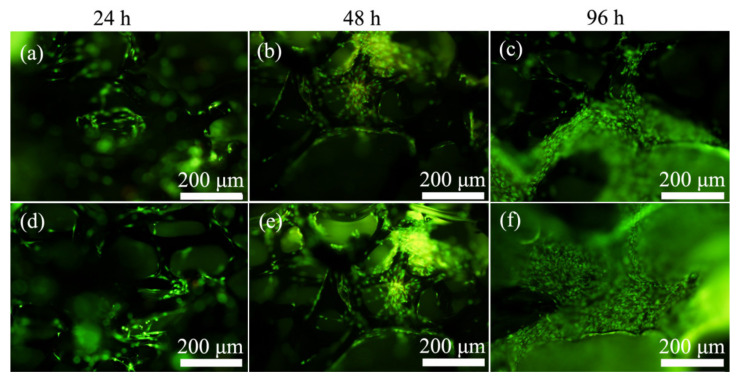
Fluorescence microscopy images of AO/EB double staining assay of MC3T3-E1 cells after co-culture with 3D rGO/PPY (**a**–**c**) and 3D rGO/PPY/Sr (**d**–**f**) composite scaffolds at 24, 48 and 96 h, respectively.

**Figure 4 molecules-26-04451-f004:**
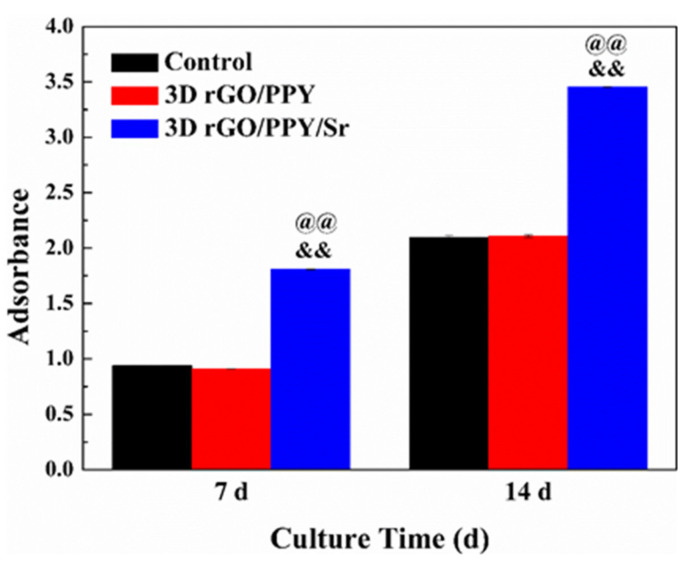
Results of the ALP assay. (&& *p* < 0.05 vs. control group, @@ *p* < 0.05 vs. 3D rGO/PPY group).

**Figure 5 molecules-26-04451-f005:**
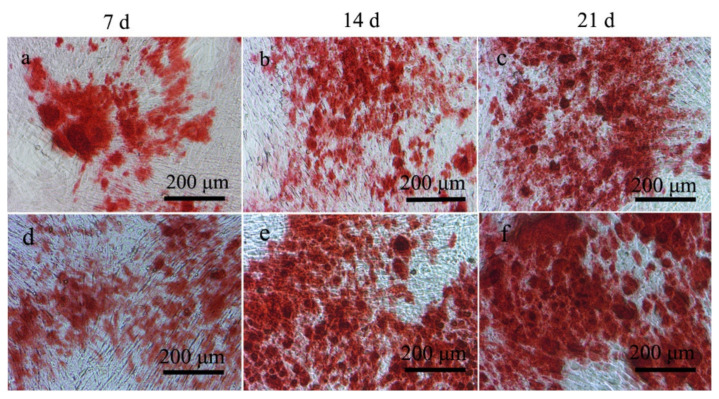
Results of the alizarin red staining assay for MC3T3-E1 cells after co-culturing with 3D rGO/PPY (**a**–**c**) and 3D rGO/PPY/Sr (**d**–**f**) composite scaffolds at 7, 14, and 21 d, respectively.

**Figure 6 molecules-26-04451-f006:**
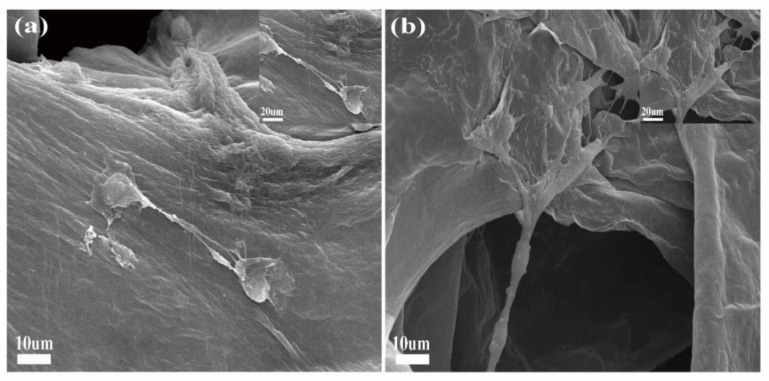
SEM images of MC3T3-E1 on the surfaces of 3D rGO/PPY/Sr scaffolds at 48 (**a**) and 96 h (**b**), respectively.

**Figure 7 molecules-26-04451-f007:**
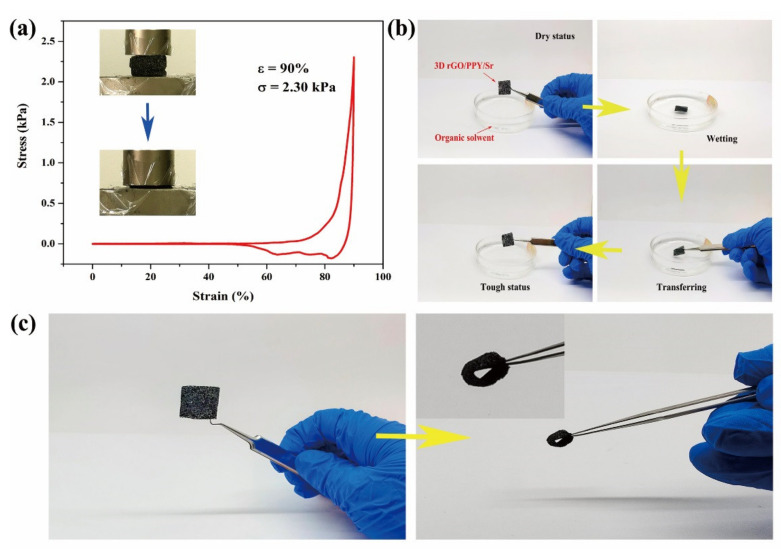
(**a**) Comparative mechanical compression test of 3D rGO/PPY/Sr scaffold with a maximum strain of up to 90%. Insets are the snapshots during compression. (**b**) Validation of mechanical robustness of 3D rGO/PPY/Sr scaffold during operation in solvent conditions. (**c**) Optical image of prepared 3D rGO/PPY/Sr scaffold with O-ring.

**Figure 8 molecules-26-04451-f008:**
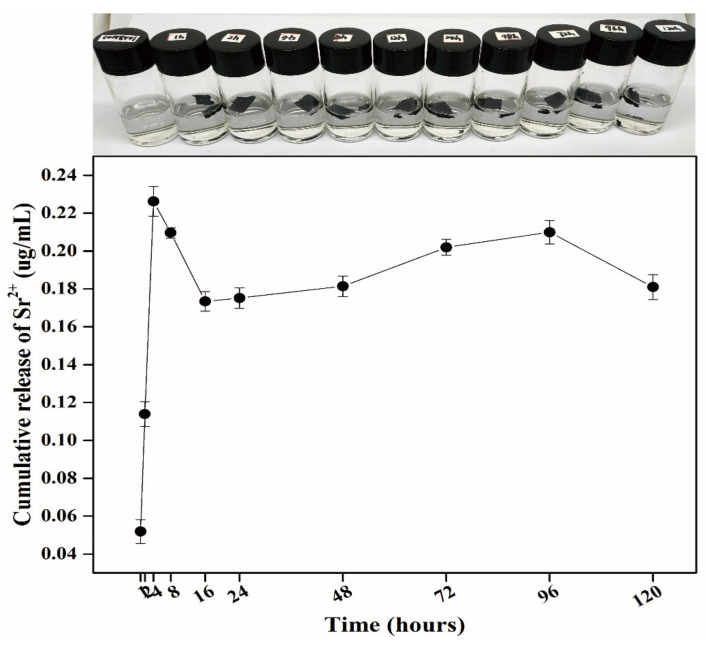
Optical image of 3D rGO/PPY/Sr scaffolds were immersed in 5 mL of PBS (the first row). Release of the Sr ions from the 3D rGO/PPY/Sr (the second row) scaffolds.

## Data Availability

The datasets generated during and/or analysed during the current study are available from the corresponding author on reasonable request.

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
