# Peer review of "Oxygen Plasma Technology-Assisted Preparation of Three-Dimensional Reduced Graphene Oxide/Polypyrrole/Strontium Composite Scaffold for Repair of Bone Defects Caused by Osteoporosis"

_molecules, 2021, doi:10.3390/molecules26154451_

Round 1

Reviewer 1 Report

The present manuscript described a type of porous scaffolds for bone tissue engineering. The scaffolds were made of graphene oxide/polypyrrole and the authors modified the scaffold with strontium (Sr) by oxygen plasma treatment. The strontium (Sr) is supposed to increase the proliferation and osteogenesis of seeded osteoblasts. The authors did a series of experiment to characterise the scaffold and evaluate the cells responses in the scaffolds. There are several concerns from the reviewer.

  1. How stable is the Sr in the scaffold? An experiment showing the release of Sr in a biological buffer should be added.
  2. The interpretation of FTIR result is not clear. I couldn’t find the peaks for Sr in the figure.
  3. Gene expression analysis of osteogenic markers is needed. They can be correlated to the Sr release profile to indicate how Sr influence the osteogenesis.
  4. SEM images for cell seeding in the scaffold are needed.
  5. Figure 3a looks like at a different magnification with the other images.
  6. Why graphene oxide/polypyrrole is used? What is the advantage to use this composite?
  7. Please carefully check the typos and mistakes in the manuscript before submission. Even in the abstract, many points need to be corrected, e.g. “a promising potentials”, “Abstract: The repairs of”(“The” should not be bold), “full name of AO/EB”….

Author Response

Dear Reviewer, 
     We would like to thank you for thoroughly reviewing our manuscript and making many thoughtful comments. We have added significant new data, described in attachment, and revised the manuscript to address your comments. 

Reviewer 2 Report

In the manuscript the authors describe the characterisation and in vitro performance of a 3D reduced graphene oxide/polypyrrole/strontium composite for use in the treatment of bone defects due to a diseased condition. While I find the study interesting, authors to address the following comments.

Minor

  1. The grammar in the following phrases and sentences thereof must be corrected.
  • Page 1 Line 11 “The repairs of bone defects…”
  • Page 1 Line 18 “that the 3D rGO/PPY/Sr has the good biocompatibility,”
  • Page Lines 21-24 Good biocompatibility, osteoconduction ……engineering”
  • Page2 Lines 90  :”And then.”
  • Page 5 line 188 “This might related to the modification of Sr,”
  • Line 248 “Meanwhile, attribute to the three-dimensional“

Major

  1. In the paper fabrication of scaffolds via the oxygen plasma-assisted strategy is proposed. I believe that this is also reflective of the novelty of the study. However, there is no background  provided on the oxygen plasma-assisted strategy. Authors should include this background in the manuscript  within in the context of the study and state the novelty of they study.
  2. The mechanical properties of the fabricated scaffolds were not discussed. Authors to possibly include this information in their discussion.
  3. Line 86. Materials should be included in the manuscript.
  4. Page 5 lines 165-169: Authors need to elaborate/ be more specific on the characterisation given.
  • For example “a large wide wave peak appears at 2θ=15-29.4°” – this is normally referred to as a indicative of amorphous phases present.
  • For example “ the peak at 2θ=11.32°, 19.6°, 44.34°, were correspond to strontium salt” Specify that these are the crystalline phases that correspond to these 2theta values”
  • For example “ In FTIR spectral (Figure. 1j), we found that the peaks corresponding to the oxygen-containing functional group of the 3D rGO/PPY/Sr were weakened” Please state which wavenumber(s) and molecular vibration(s) is/are you are referring to and clarify why the molecular vibration band has lost intensity? A reference to support this observation should be included.
  1. Please comment: can the proposed scaffolds conform to the spatial/ shape of the specified bone defect?
  2. Comment: has the biodegradation of the scaffolds been investigated?

Author Response

(The authors gave the same response as above.)

Round 2

Reviewer 1 Report

The authors have addressed all the issues.

Reviewer 2 Report

All comments have been addressed.